# Athlete Identity and Mental Health of Student Athletes during COVID-19

**DOI:** 10.3390/ijerph192417062

**Published:** 2022-12-19

**Authors:** Katherine Antoniak, Clea Tucker, Katherine Rizzone, Tishya A. L. Wren, Bianca Edison

**Affiliations:** 1Children’s Hospital Los Angeles, Los Angeles, CA 90027, USA; 2School of Kinesiology, California State University Los Angeles, Los Angeles, CA 90032, USA; 3Department of Orthopaedics, University of Rochester Medical Center, Rochester, NY 15618, USA; 4Keck School of Medicine, University of Southern California, Los Angeles, CA 90033, USA

**Keywords:** COVID-19, athlete, athletic identity, mental health, adolescent and young adult, youth, health

## Abstract

The purpose of our study is to evaluate athletic identity (AI) and mental health measures of youth and young adult athletes during the COVID-19 pandemic. This cross-sectional study recruited athletes aged 11–25 years from universities, high schools, and middle schools in California and New York. Participants were emailed a link to an anonymous, cross-sectional electronic survey. The measure included the athletic identity measurement scale (AIMS), the Patient Health Questionnaire-4 (PHQ-4), and demographic variables. Chi-square, Fisher’s Exact Test, and linear regression were used to examine the relationships between AI, symptoms of anxiety, and symptoms of depression by age, gender, and race. The survey was completed by 653 participants. AI was stratified by tertiary percentiles. The odds of positively scoring for symptoms of anxiety were 60% higher for participants in college compared with high school (OR: 1.60, 95% CI: [1.09, 2.35]). Conversely, the odds of scoring positively for symptoms of depression were 68% higher for participants in high school compared to college (OR: 1.68, 95% CI: [1.09, 2.59]). The odds of scoring positively for symptoms of depression were higher for athletes who scored as high AI, compared to those who scored as moderate (OR: 1.72, 95% CI: [1.11, 2.68]) or low (OR: 1.93, 95% CI: [1.20, 3.12]). The odds of scoring positively for symptoms of anxiety on the PHQ-4 were 3.2 times higher for participants who identified as female (OR: 3.19, 95% CI: [2.31, 4.41]), and the odds of scoring positively for symptoms of depression were 2.4 times higher for participants who identified as female (OR: 2.35, 95% CI: [1.56, 3.54]). Female athletes experienced symptoms of depression and anxiety at significantly higher rates than male athletes during the COVID-19 pandemic. High school students experienced fewer symptoms of anxiety, but greater symptoms of depression as compared to the collegiate group, while college students experienced greater odds of symptoms of anxiety. Athletes in the high AI group were more likely to report symptoms of depression than moderate or low identity groups. Female athletes reported lower AI than male athletes, but still had greater symptoms of anxiety and depression.

## 1. Introduction

The 11 March 2020 will indelibly be marked in history as a globally-memorable date, as on that day, the World Health Organization declared the severe acute respiratory syndrome coronavirus 2 (SARS-CoV-2) as a pandemic [1]. The onset of the COVID-19 pandemic shut down athletic team practices and competitive sporting events across the globe and threatened the mental health and livelihood of individuals and communities globally. In an unprecedented move in 2020, major athletic governing bodies such as the National Collegiate Athletic Association (NCAA), the National Basketball Association (NBA), and Major League Baseball (MLB) suspended athletic events and practices in response to the novel virus. Several league drafts and the 2020 Tokyo winter Olympics were postponed, and college and high school senior athletes were unable to complete their final season [2]. The level of uncertainty surrounding athletes’ futures had been experienced at a magnitude never seen before. Uncertainty of the future, abrupt shifts of daily activities to a virtual platform (including school and physical activity), social isolation, distancing and masking mandates, financial stressors, and fear regarding that uncertainty fueled a surge of mental health ailments [3,4]. 

The ability to participate in sports has been well-researched as being integral to the well-being of athletes, as engagement is deeply intertwined with personal identity [5,6,7,8,9,10,11,12,13,14,15,16,17]. Identity can be defined as role-based self-conceptions with perceived expectations related to the performance of a social role [8]. This can extend to include athletes, though current research has taken a more nuanced framework that is mirrored in this paper [9]. Rather than strictly identify Athletic Identity (AI) based on roles, current AI research has considered AI as a “complex cultural construction” [9]. 

Brewer, Van Raalte, and Linder’s research included some of the first comprehensive efforts to conceptualize and systematically study athletic identity [9]. They defined AI as the degree to which an individual identifies with the athletic role, or the degree to which one devotes special attention to sport, relative to other activities in life [9]. In 1997, Wiechmen et al. published a novel study that examined the role of athletic identity in relation to sports injuries and mood disturbance [10]. Predictors of high AI scores included demographic background (gender and ethnicity) and years of sports participation. These studies served as the foundational basis for this field of study with subsequent research investigating the relationship between AI and chronic pain [11], relationship to retirement from sports [12], and burnout [13]. A 2001 study by Barber et al. showed that sports participation generally resulted in long term positive academic and occupational success [14]. Similarly, Phoenix et al. asserted that athletes’ self-worth positively correlated with healthy aging expectations; however, a strong athletic identity alone negatively correlated with views on aging and the future self [15]. Athletic identity can also be a critical predictor for injuries, and even student athlete academic success [16,17]. High athletic identity can be associated with positive health and performance outcomes, but can also be associated with negative sequelae including overtraining, use of performance enhancing drugs, and psychological health disturbances [18]. A study examining female athletes showed that higher obsessiveness and athletic identity correlated to higher rates of burnout [17]. These studies highlight that AI is an important factor when considering the mental and physical health of athletes, as well as academic success. 

Mental health for elite and student athletes is a well-researched topic. Former student athletes, those who used to play sports and now do not, report a prevalence of depressive symptoms and symptoms of anxiety at 10.4% and 16.2%, respectively. Current athletes report rates of anxiety and depression-related symptoms as high as 34.0%. These rates are lower than the general non-athlete population [19]. Approximately 23.7% of college athletes reported clinically significant depressive symptoms over a 5-year period using a validated depression screening tool (the Center for Epidemiological Studies Depression scale) in the largest published study to date in the sports medicine literature [20]. Many risk factors exist for mental health disturbances in athletes, which include personality traits such as perfectionism, sexuality and gender issues, hazing and bullying, sexual misconduct, injury and illness, and transition from sport [21]. Injuries, and the inability to participate in sport, have often been cited as factors associated with worsening mental health outcomes in athletes [22]. 

The COVID-19 pandemic ushered in a number of new mental health studies that focused on athletes. These have included a study that examined the relationship between AI, mental health, and social support; finding that AI was negatively associated with depression and helplessness among varying age groups [23]. Another major study included a scoping review that identified variables most often used and tested in research studies on depression. This study found that only 5% of observations accounted for macro-level variables, such as ethnicity, meaning that most studies on depression fail to look at its relationship with ethnicity and demographic variables [24]. Both research groups recognized the need for more research and intersectional approaches that explore the relationships between AI, demographic variables, and symptoms of depression or anxiety.

The findings of the aforementioned mental health studies are troubling in the context of the COVID-19 pandemic, where worsened mental health outcomes have been seen across the United States and world [25]. It is important to be able to properly recognize these risk factors, as well as appropriately manage mental health outcomes in athlete populations. The purpose of our study was to examine athletic identity and mental health in young athletes during the COVID-19 pandemic. Given the dramatic and historic changes in the sports landscape, we hoped to examine AI and mental health during the pandemic to better understand the impact COVID-19 may have had on the lives of youth and young adult student athletes. We hypothesized that higher self-rated levels of AI were associated with higher depression and anxiety symptoms in the context of the COVID-19 pandemic. Our secondary hypothesis was that female and collegiate athletes would experience greater symptoms of anxiety and depression than male athletes, or athletes at other school levels. The results of this study could be utilized by other investigators, coaches, and academic personnel interacting with student-athletes. 

## 2. Materials and Methods

This study involved data analysis of a cross-sectional survey conducted between April 2020 and October 2020. Athletes aged 11–25 years old were recruited from universities, high schools, and middle schools in Central New York and Southern California. Our group wanted to assess potential differences between periods in which student athletes tend to enter, participate in, and transition out of sports, which involves middle school, high school, and college, respectively [26]. Those academic levels of training translate to the ages of 11–25 years. Athletes were included up until the age of 25 because there were college students within this age range in our sample. The study was approved by the authors’ Institutional Review Boards. Participants were contacted through university sports programs and local youth athletic organizations. Participants were emailed a link to an anonymous, cross-sectional electronic survey. Research Electronic Data Capture (REDCap) was utilized to develop and administer the survey and measures, which provided a secure, web-based software platform designed to support data capture for research studies, providing procedures for data integration and interoperability with external sources [27].

### 2.1. Measures

The survey was developed by a sports medicine team that included a primary care sports medicine physician, athletic trainers, athletes, and coaches. Wherever possible, we included clinically validated instrument measures. After initial construction, it was reviewed and edited by the study team. Edits were made iteratively based on feedback from initial testing with student athletes to ensure the survey captured the intended data, and did not incite unnecessary survey fatigue. The survey required approximately 15 min of an individual student’s time to complete. 

### 2.2. Athletic Identity Measurement Scale (AIMS)

Athletic identity was determined based on the participant’s responses to a modified validated tool with ratings ranging from strongly disagree (score = 1) to strongly agree (score = 7). Initially presented as a unidimensional construct involving a 10-item questionnaire, the AIMS tool has been reconceptualized into a seven-item questionnaire that consists of three dimensions: social identity, negative affectivity, and exclusivity [9]. Social identity involves the extent that an individual connects to and occupies an athlete’s role. Exclusivity within AIMS describes the level to which an individual’s identity and self-worth are constructed by the performance within an athletic role. Negative affectivity represents the extent to which one experiences negative sequelae in response to undesirable outcomes within the athletic role [28]. The AIMS instrument has been validated in both collegiate and pediatric populations [29,30]. Based on the total score summing all items, athlete identity was stratified by tertiary percentiles into low (total 0–34), medium (total 35–40), or high (total 41–49) athletic identity groups.

### 2.3. Modified AIMS Questionnaire

“Please rate the following as it pertains to you NOW:”

I consider myself an AthleteI have many goals related to sportMost of my friends are athletesSport is the most important thing in my lifeI spend more time thinking about my sport than anything elseI feel bad about myself when I do poorly in my sportI am very depressed that I currently cannot compete in my sport due to COVID-19/Coronavirus restrictions

### 2.4. Patient Health Questionnaire-4 (PHQ-4)

To assess symptoms of depression and anxiety, the research team utilized the PHQ-4, a validated four-item measure that elucidates two items unique to anxiety and two items targeting depressive symptoms [31]. The first two questions in this section focus on anxiety, utilizing a Likert scale (0 = not at all, 1 = several days, 2 = more than half the days, 3 = nearly every day), while the subsequent two questions ask about symptoms of depression using the same scale. Participants were asked “Over the LAST 2 WEEKS, how often have you been bothered by the following problems:”

A1. Feeling nervous, anxious or on edge

A2. Not being able to stop or control worrying

D1. Little interest or pleasure in doing things

D2. Feeling down, depressed or hopeless

Participants were considered to have symptoms of anxiety if the sum of their scores for A1 + A2 was ≥ 3, and symptoms of depression if the sum of their scores for D1 + D2 was ≥ 3 [31].

The maximum score is 12 points. This measure has been utilized in prior studies in youth populations to include college students and pediatric populations [31,32,33]. Scores are rated as normal (0–2), mild (3–5), moderate (6–8), and severe (9–12).

Participants were asked demographic questions including date of birth, year in school, sex, gender, race, ethnicity, geographic location, sport, and athletic status. For gender, participants were asked if they identified as “male,” “female,” “non-binary,” or “preferred not to say”.

### 2.5. Analytics Plan

Descriptive statistics showed categorical variables as proportions with percentages and continuous variables as means with standard deviations. Chi-square tests or Fisher’s Exact Test were used to examine the relationships between AI, symptoms of anxiety and depression, and demographic variables. Significant relationships were further explored using binary logistic regression or binary ordered logistic regression. The demographic variables considered included gender, race, ethnicity, school level (middle school, high school, or college/graduate school), and geographic location. Statistical analysis was performed using StataIC 14 (StataCorp. 2015. Stata Statistical Software: Release 14. College Station, TX, USA: StataCorp LP). A *p*-value threshold of alpha < 0.05 was used to ascertain significance of statistical tests.

## 3. Results

The survey was distributed to 797 participants including 144 (18.0%) who did not return the survey. Of the 144 respondents who were not included in the final sample, only 15 were excluded due to missing data, and the remaining 129 were excluded because they did not return the survey, yielding 653 participants that were used in the final analysis (Table 1). Of those participants, 44.0% were male, 55.6% were female, and 0.4% were nonbinary (two participants) or did not indicate their gender (one participant). Due to the small number of individuals with nonbinary or unspecified gender identity, subsequent analyses involving gender were performed using only males and females. None of the respondents indicated a self-identification as transgender. White-identifying students were 59.3% (387/653) of our sample, 10.4% identified (68/653) as Black, 7.0% (46/653) as Asian, 21.8% (142/653) as other or mixed race, and 1.5% (10/653) preferred not to indicate a race category. Latinx-identifying participants made up 26.8% of our sample. Participants were primarily from California at 48.5% (317/653) and New York at 41.5% (271/653). College students were 74.7% (488/653) of our sample, high school students were 20.5% (134/653), and middle school students were 4.5% (31/653).

The mean athletic identity score was 37.5 (SD 6.7). For the athletic identity groups, 36.6% (239/653) of participants were scored as high athletic identity, 34.6% (226/653) as moderate athletic identity, and 28.5% (18/653) as low athletic identity. AI differed significantly by gender, race, ethnicity, school level, and geographic location (all *p* < 0.04, Table 2 and Table 3).

Symptoms of anxiety and depression were observed in 53% (348/653) and 21% (138/653) of participants, respectively. Symptoms of anxiety and depression from PHQ-4 scores differed based on gender (both *p* < 0.001) and school level (both *p* < 0.05). Anxiety symptoms were also related to race (*p* = 0.03), where those participants identifying as Asian reported lower levels of symptoms of anxiety, as compared to other racial identities (OR: 0.44, 95% CI: [0.23, 0.82]). Symptoms of depression were related to athletic identity (*p* = 0.008) (Table 4). Both the odds of the symptoms of anxiety (OR: 3.19, 95% CI: [2.31, 4.41]) and depressive symptoms (OR: 2.35, 95% CI: [1.56, 3.54]) were greater in females compared with males (Table 3). The odds of the symptoms of anxiety were less in high schoolers compared to college students (OR: 0.62, 95% CI: [0.43, 0.92]), but the odds of the symptoms of depression were greater for high schoolers compared to college students (OR: 1.68, 95% CI: [1.09, 2.59]. Consequently, the odds of having symptoms of anxiety were 60% higher for participants in college compared with high school (OR: 1.60, 95% CI: [1.09, 2.35]). In addition, the odds of having symptoms of depression were greater in participants with high AI, compared to low AI (OR 1.93, 95% CI: [1.20, 3.12]) or medium AI (OR: 1.72, 95% CI: [1.11, 2.68]).

## 4. Discussion

Key findings of this study included that the odds of high AI were greater in participants who were male compared to female, Latinx compared to non-Latinx, in secondary school compared to college, and living in California compared to New York or other locations. Females experienced greater odds of symptoms of depression and symptoms of anxiety than males. High school students experienced higher odds of symptoms of depression compared to other school levels, but college students had greater odds of symptoms of anxiety than other school levels. Athletes in the highest tertile of athletic identity were more likely to report depression symptoms than those with lower athletic identity, but females had higher odds of anxiety and depression symptoms than males despite having lower levels of athletic identity.

The results of our study revealed concerning findings during the COVID-19 pandemic in reported symptoms of depression and anxiety among student athletes at the high school level, athletes with a high AI, and female athletes. High school students in our study had 74.0% increased odds of reporting symptoms of depression compared to college or middle school students. Previous research suggests that depression in high school students more than doubles the odds they will drop out of high school [18]. In contrast, college students in our study experienced a 47.0% increased odds of reporting anxiety symptoms compared to middle and high school students. While our results are congruent with previous research at the middle school level, and related to gender, they differ from other previous research. Our results showed that high school students had greater symptoms of depression than college students. Other research has reported that depression and anxiety increase as students get older [31,32]. However, prior research has suggested that up to 75% of lifetime cases among adults are derived from adolescent-onset disorder [33]. The COVID-19 pandemic could also serve as a possible cause for this change in symptoms of depression in high school students as COVID-19 dramatically altered the environment for these teens. Both anxiety and suicide rates were dramatically increasing before the COVID-19 pandemic for US undergraduate students [34]. Other research conducted on the general collegiate population during the COVID-19 pandemic has also reported increased rates of anxiety and depression [35]. Suicide, which is associated with depression [36], is the second leading cause of death in young adults in the United States [37]. Athletic participation can provide one with a strong group membership, and support network to face hardships and trauma. It can also create opportunities for stress, anxiety and risk-taking behavior [16]. Research has shown that depression can increase the risk of suicide for an individual, but the interplay between sport participation, depression, and suicide is more complex and not fully understood.

Athletes who scored in the highest AI category had 72–93% higher odds of having symptoms of depression compared to the moderate or low AI groups. AI is the degree to which one devotes special attention to sport compared to other activities and defines themselves by sport participation. The finding that student athletes with higher AI scores had higher symptoms of depression is corroborated by a body of literature that suggests athletes experience reduced mental health outcomes when they are unable to participate in their respective sports, particularly when their levels of athletic exclusivity identity are high [22,38,39]. Future research should explore causal relationships for these findings.

The greatest disparities seen in this study involved female athletes, who had 3.2 times the odds of reporting symptoms of anxiety, and 2.4 times the odds of reporting symptoms of depression than their male counterparts. Sadly, the disparities in mental health by gender are well known, with female athletes experiencing eating disorders, anxiety, and depression at higher rates than male athletes [20,31,40]. It has been reported that female athletes have almost twice the risk of having depression symptoms as male athletes [20]. Other research involving athletes during the COVID-19 pandemic also supports the notion that female athletes have experienced higher symptoms of anxiety and depression than males [41]. The severity of increased risk of symptoms of depression and anxiety in an already higher-risk population cannot be overstated, and further clinical screening for depression and anxiety is needed for this population.

When assessing race and ethnicity differences, self-reported symptoms of anxiety were found to be lower in Asian athletes, as compared to other racial groups. However, this should not be minimized; in total, 35.0% of participants who identified as Asian were found to have symptoms of anxiety, compared to 56% of athletes identifying as White and 47.0% of athletes identifying as Black or African American. This finding supports other prior research during this time; in a study of 7143 Chinese collegiate students, over 23.0% met criteria for anxiety [35]. Reported symptoms of anxiety and depression were roughly equal between Latinx and non-Latinx individuals. White participants reported the highest percentage of symptoms of anxiety compared to Black participants, Asian participants or other groups in the study. These findings are echoed in the larger body of literature on race, ethnicity and mental health. A prominent 2020 study also reported that Whites reported higher levels of general anxiety, and that Asian participants reported the lowest levels of anxiety of any group included. This paper also suggested these disparities in anxiety could be due to biases in language used in diagnostic screening tools, which are largely validated on White cohorts [42]. The implications of our study, namely the negative associations between symptoms of depression and anxiety with race, ethnicity, school level, gender and geography, echo those seen in other publications on athlete mental health since the onset of the COVID-19 pandemic [23,24]. Namely, sports are an integral part of many students’ lives and identities. While there is a large body of research on sports and mental health, much of this fails to focus on macro factors, such as ethnicity or race [24]. We strongly recommend that future research endeavors in this space take these intersectional and demographic factors into consideration.

There were several limitations to our study. First, this was a cross-sectional survey study, and thus we did not have any data prior to the pandemic to compare these variables before and after the COVID-19 pandemic. In addition, given the nature of distribution of the survey, determination of response rate was difficult to ascertain. Additionally, 18% of our original sample was not included in the final analysis. Of these, 129 (16.2%) did not return the survey and 15 (1.9%) had missing data. This may have introduced a small amount of bias into our study. There were no prominent patterns of missing data, but survey responses are generally lower on web-based surveys compared to other methods [43]. Possible reasons for participants not completing their surveys could have included disinterest, survey fatigue, or not wanting to answer certain questions due to privacy concerns. Additionally, data was only collected for date of birth to ascertain age in the middle and high school students, but was not collected in the college students, so age was not included as a demographic variable. Regarding the subject group, we had representation from California and New York, which may not be representative of the youth and young adult athlete population in the broader United States. Middle School students only represented 4.5% of the population, so we would want to study this group of athletes in larger numbers to look at trends. Our study found some statistically significant differences between racial and ethnic groups, but these nuances need to be studied in more robust detail; in particular, more disaggregated data within these categorizations or groupings need to be analyzed. In addition, we had such a small number of individuals who identified as non-binary for gender, that we could not adequately study this group. Research is lacking for athletes who are non-binary and part of the LGBTQIA+ communities, particularly transgender athletes, so future studies need to be conducted and should purposefully ensure more inclusion of a more diverse population.

## 5. Conclusions

This research study sought to examine the impact of the COVID-19 pandemic on the emotional and mental health of youth and collegiate athletes. Our study revealed that female athletes had lower athletic identity and experienced symptoms of depression and anxiety at significantly higher rates than male athletes during the COVID-19 pandemic. High school students experienced less frequently reported symptoms of anxiety, but higher reported symptoms of depression compared with college students. Athletes with high athletic identity were more likely to report symptoms of depression than those with moderate or low athletic identity. The mental state of an athlete, and how they internalize and process expectations, can have a large impact on their trajectory and future approach to physical activity. As the COVID-19 pandemic continues, amidst changes in the landscape of athletics to include budget cuts, staffing, and public health measures, the extent to which an athlete identifies with their sport, access to other outlets, and emotional well-being will continue to be critical aspects to monitor and study so proper resources can be made available to all athletes.

## Figures and Tables

**Table 1 ijerph-19-17062-t001:** Participant characteristics.

Characteristic	n/N (%)
Gender	
Male	287/653 (44.0%)
Female	363/653 (55.6%)
Non-Binary	2/653 (0.3%)
N/A	1/653 (0.2%)
Race	
White	387/653 (59.3%)
Black	68/653 (10.4%)
Asian	46/653 (7.0%)
Other or Mixed	142/653 (21.8%)
Not specified	10/653 (1.5%)
Ethnicity	
Hispanic or LatinX	172/653 (26.8%)
Not Hispanic or LatinX	471/653 (73.3%)
School level	
Middle school	31/653 (4.8%)
High school	134/653 (20.5)
College	488/233 (74.7%)
Location	
California	317/653 (48.5%)
New York	271/653 (41.5%)
Other	65/653 (10.0%)
Athletic Identity (AI)	
High	239/653 (36.6%)
Medium	226/653 (34.6%)
Low	188/653 (28.8%)

**Table 2 ijerph-19-17062-t002:** Frequency of athletic identity tertiles by demographic variables.

	Athlete Identity			Chi-Square
	Low	Medium	High	*p*-Value
All Participants	188/653 (29%)	226/653 (35%)	239/653 (37%)	
Gender				**0.04**
Male	**68/287 (24%)**	**103/287 (36%)**	**116/287 (40%)**	
Female	**118/363 (33%)**	**123/363 (34%)**	**122/363 (34%)**	
Race				**0.002**
White	**122/387 (32%)**	**141/387 (36%)**	**124/387 (32%)**	
Black	**18/68 (26%)**	**27/68 (40%)**	**23/68 (34%)**	
Asian	**15/46 (33%)**	**17/46 (37%)**	**14/46 (30%)**	
Other or Mixed	**28/142 (20%)**	**39/142 (27%)**	**75/142 (53%)**	
Ethnicity				**<0.001**
Not Hispanic or Latinx	**146/471 (31%)**	**174/471 (37%)**	**151/471 (32%)**	
Hispanic or Latinx	**39/172 (23%)**	**49/172 (28%)**	**84/172 (49%)**	
School level				**0.03**
Middle school	**6/31 (19%)**	**7/31 (23%)**	**18/31 (58%)**	
High school	**34/134 (25%)**	**42/134 (31%)**	**58/134 (43%)**	
College	**148/488 (30%)**	**177/488 (36%)**	**163/488 (33%)**	
Location				**<0.001**
New York	**101/271 (37%)**	**110/271 (41%)**	**60/271 (22%)**	
California	**60/317 (19%)**	**93/317 (29%)**	**164/317 (52%)**	
Other	**27/65 (42%)**	**23/65 (35%)**	**15/64 (23%)**	

Bolded items were found to have met the threshold of *p*-value < 0.05.

**Table 3 ijerph-19-17062-t003:** Associations between athletic identity, symptoms of anxiety, and symptoms of depression and demographic variables.

	Athletic Identity		Anxiety		Depression	
	OR [95% CI]	*p*	OR [95% CI]	*p*	OR [95% CI]	*p*
Female vs. Male	**0.70 [0.53, 0.93]**	**0.02**	**3.19 [2.31, 4.41]**	**<0.001**	**2.35 [1.56, 3.54]**	**<0.001**
Race						
Other/Mixed vs. White, Black, Asian	**2.20 [1.54, 3.14]**	**<0.001**	1.09 [0.75, 1.58]	0.66	1.17 [0.75, 1.82]	0.49
Asian vs. White, Black, Other/Mixed	0.78 [0.45, 1.35]	0.38	**0.44 [0.24, 0.83]**	**0.01**	0.65 [0.29, 1.49]	0.31
Latinx	**1.83 [1.32, 2.55]**	**<0.001**	0.92 (0.65, 1.31]	0.65	0.92 [0.60, 1.42]	0.72
School level						
Middle School vs. College	**2.47 [1.20, 5.08]**	**0.01**	0.96 [0.46, 2.00]	0.92	0.44 [0.13, 1.47]	0.18
High School vs. College	**1.42 [0.99, 2.03]**	**0.05**	**0.62 [0.43, 0.92]**	**0.02**	**1.68 [1.09, 2.59]**	**0.02**
High School vs. Middle School	0.57 [0.27, 1.24]	0.16	0.65 [0.30, 1.42]	0.28	**3.83 [1.10, 13.34]**	**0.04**
Location						
California vs. New York	**3.16 [2.32, 4.31]**	**<0.001**	1.01 [0.73, 1.40]	0.96	1.31 [0.87, 1.96]	0.20
California vs. Other	**3.47 [2.09, 5.74]**	**<0.001**	0.85 [0.50, 1.46]	0.56	0.75 [0.41, 1.38]	0.36
High vs. Medium/Low Athletic Identity	---	---	1.22 [0.89, 1.69]	0.21	**1.81 [1.24, 2.65]**	**0.002**

Bolded items were found to have met the threshold of *p*-value < 0.05.

**Table 4 ijerph-19-17062-t004:** Frequency of symptoms of anxiety and depression by demographic variables.

	Anxiety	*p*-Value, Chi-Square	Depression	*p*-Value, Chi-Square
All Participants	348/653 (53%)		138/653 (21%)	
Gender		**<0.001**		**<0.001**
Male	**108/287 (38%)**		**39/287 (14%)**	
Female	**239/363 (66%)**		**98/363 (27%)**	
Race		**0.03**		0.65
White	**218/387 (56%)**		80/387 (21%)	
Black	**32/68 (47%)**		16/68 (24%)	
Asian	**16/46 (35%)**		7/46 (15%)	
Other or Mixed	**78/142 (55%)**		33/142 (23%)	
Ethnicity		0.65		0.72
Not Hispanic or Latinx	256/471 (54%)		102/471 (22%)	
Hispanic or Latinx	90/172 (52%)		35/172 (20%)	
School level		**0.05**		**0.02**
Middle school	**17/31 (55%)**		**3/31 (10%)**	
High school	**59/134 (44%)**		**39/134 (29%)**	
College	**272/488 (56%)**		**96/488 (20%)**	
Location		0.83		0.17
New York	143/271 (53%)		49/271 (18%)	
California	168/317 (53%)		71/317 (22%)	
Other	37/65 (57%)		18/65 (28%)	
Athlete identity		0.46		**0.008**
Low	97/188 (52%)		**31/188 (16%)**	
Medium	116/226 (51%)		**41/226 (18%)**	
High	135/239 (57%)		**66/239 (28%)**	

Bolded items were found to have met the threshold of *p*-value < 0.05.

## Data Availability

The data presented in this study are available on request from the corresponding author. The data are not publicly available due to hospital policy.

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
