# Peer review of "Athlete Identity and Mental Health of Student Athletes during COVID-19"

_ijerph, 2022, doi:10.3390/ijerph192417062_

Round 1
Reviewer 1 Report
Thank you for the opportunity to review this paper on athlete identity and mental health for young athletes during the COVID-19 pandemic. This manuscript has some important strengths, and this discussion of how COVID impacted student athletes is important to have as we return to 'post-COVID' athletic activities. Overall, I think many will find this manuscript interesting and the large, diverse sample provides a great opportunity to explore these questions.
I have some comments which I feel would be important for improving this manuscript before it is accepted for publication. Most importantly, I would advise authors to be cautious about their use of language and ensuring their conclusions are grounded in their findings and the measures they used.
Major comments:
- I would caution the authors to be conscious of their use of terminology around anxiety and depression. The PHQ-4, while a useful screening tool, is by no means sufficient for diagnosis. Throughout the manuscript, the authors' language suggests that those participants meeting screening criteria on this measure 'have' anxiety and depression. I would request that the authors be cautious not to over state this measure's diagnostic capacity by suggesting that the PHQ-4 alone can show 'rates' of anxiety and depression. One suggestion to mitigate this could be to replace 'rates of anxiety and depression' with 'symptoms of anxiety and depression'. This is a recurrent issue throughout the abstract, results, discussion, and conclusion.
- 18% seems to be a very high portion of incomplete data excluded. I would request that the authors discuss this further, such as in their limitations section, considering possible explanations for this high portion of incomplete data. Where there any patterns to whose data was incomplete? Are there any possible explanations for this, e.g., survey fatigue or particular questions that were not answered?
- The authors identify their survey respondents' gender as 'male' or 'female'. Were these the exact terms used in the survey? Was there any indication of whether these groups included trans individuals? Given the additional stressors faced by trans athletes, this seems important to discuss.
- The authors do not discuss age. I would suggest that they please consider including age in their demographics. While school level is useful for delineating some of how these relationships present over these age groups, age would also be useful for exploring this given the wide developmental gap between 14 and 18 year olds in high school, for example.
- Was there any indication of the impact COVID had on individuals, such as a report on whether their specific activities were disrupted by COVID? I am curious about this as the authors appear to state in the discussion that loss of activity was the cause of the increased positive depression screen among athletes with higher AI in paragraph 2 of the discussion. At present this seems to over-reach the conclusions we could draw from the presented analysis, which did not take a mechanistic approach or assess disruption. I agree that the disruption to activity is a plausible and compelling explanation for the present findings, but would advise the authors pose this as a possible explanation to avoid this coming across as a causative claim without empirical basis.
- Similarly to my previous comment, the authors state that their findings 'highlight that those athletes may not have other alternative outlets'. On what basis is this claim made? Was coping assessed?
- It is clear from both the present findings and past research that particular attention to the mental health of female athletes is warranted. However, the statement in line 306 that clinical intervention is critical again feels disproportionate to the present findings as we are unable to accurately determine from PHQ-4 results how many of these participants would meet clinical diagnostic threshold.
- In line 308, the authors suggest that the lower AI of female participants along with higher positive screens for depression suggests that importance of proper identification and targeting of treatment modalities. The assumption underlying this statement is that the mechanisms of the positive depression screen appears different for some (those with high AI vs female athletes with lower AI). However, exploring and confirming underlying mechanisms appears to be an important missing step prior to suggesting treatment modalities will be different across populations.
- I would request that the authors include greater discussion of their findings re race and age which are currently under explored in their discussion.
Minor comments
- There appears to be an error on the first page with the abstract printed multiple times. I ask the authors please correct this and include their preferred abstract.
- The paragraph in the introduction on athletic identity could be re-worded for improved flow - at times it is unclear how the different aspects of AI link together.
- I ask the authors to please include their hypotheses.
- I ask the authors to please spell out all acronyms in full the first time they are used in the body of the text (not just the abstract).
- The final paragraph of the 'measures' section within the methodology appears to be an analytic plan or procedure. I would request the authors include an appropriate sub-heading to make this easier for readers to find.
- The authors group together respondents with non-binary identity and who did not indicate gender. While both groups will be too small to analyze, I would request that they please separate these groups and provide n for both as these are two very different populations.
- Please include cut off scores for all measures in the measures section.
- I would request the authors add a brief summary paragraph recapping their key findings to the opening of the discussion.
- I'm somewhat confused by the inclusion of line 272-273 on depression and suicide research. The previous sentence - discussing the lack of evidence on the link between sport participation, depression, and suicide - appears grounded in this manuscript. However, the statement that 'more studies are needed to understand whether there is a cause-and-effect relationship between depression and suicide or an associative one' feels out of place in this manuscript and very under-explained. Quite a lot of evidence exists that this is not a cause and effect relationship and I'm unsure how this relates to these findings. Please consider revising or removing.
- In the discussion, I would suggest the authors please start a new paragraph at line 309 to separate the discussion of findings related to race from findings related to gender.
Author Response
Please find attached our responses to your comments. Thank you for your consideration of this revised manuscript.

Reviewer 2 Report
This was an interesting study about an important issue: athletes, mental well-being, and athletic identity. However, the study had a number of serious flaws that must be addressed before being reconsidered for publication.
Abstract
- There appears to be two abstracts presented in the pdf version of the manuscript that I read. The second abstract was written much more clearly than the first abstract.
- avoid the use "one-time" survey, and instead use the correct term "cross-sectional."
- Do not begin sentences with numbers.
- presented in the Abstract and the Results section, the authors need to review the standard practices for accurately reporting ORs. The presentation of the findings should be consistent too.
- Overall, the paper needs to be reviewed for consistency. There are sections of the manuscript that appear to be written by different people with different writing styles. This has also created redundancies in content. When this occurs, the overall quality of the paper decreases.
Introduction
Line 97: The statement "The athletic culture and one’s environment can contribute to one’s identity as well as the mental health of athletes" needs a reference. Moreover, this is a broad statement that requires further description.
- It should be written as COVID not Covid.
- Line 100: What does "former athletes" refer to?
- Line 105: remove brackets and write in complete sentences
- Line 113: the study fails to communicate the justification for the study other than mere exploration. What are the outcomes associated with poor mental health among athletes? How does athletic identity moderate this? Who will use the results? The statements provided are too vague.
Line 117: remove "hope" because it suggests that the study was not firmly grounded in previous research or theory in order to generate a plausible set of research aims or hypotheses. On that point, there are no research goals or hypotheses presented despite the opportunity to do so.
- There is no theoretical framework presented. The reference for athletic identity is very dated, and comes from a conference paper. Is there no other research on the topic?
- The literature review fails to adequately discuss recently published studies about athletes and mental health.
Line 119: The last two sentences of the Introduction appear to present some of the results. These statements should be removed.
- I am unclear why a sample of 11-25 year old athletes were used. What is the justification for including this range? The literature review should have described the developmental differences. Would we consider 20-25-year old athletes "young"? The description of the sample needs to be clarified.
- Line 123: It is unclear was “retrospective data analysis of a cross-sectional study” means.
- Line 146: Please write the full names of the instruments used in the Materials section.
- Figure 1 is not properly formatted.
- Line 159: Categories for low and medium AI overlap. This needs to be revised.
- Line 163: It appears that there are two sections of PHQ-4 and PHQ4 in the Materials section.
- Line 165: please replace the term "geared" with a more scientifically precise term.
- Line 198: Include sub-heading for data analysis
- In the Methods section, there is no description of how the demographic variables were assessed
- Line 209: what was the criteria cut-off for excluding incomplete data?
- Please use the term "participants" rather than "subjects"
- Table 1 formatting problems. Further, in Table 1, it is unclear what Athlete ID means.
- Line 226: the text repeats what is in the table.
- Table 2 and Table 3 have formatting problems. It is unclear why there is bolded items or why exact p values are presented. Please review the standard practice for formatting using a particular style.
- Line 225: The presentation of ORs is not accurate and is presently inconsistently in the text. Also, please tell the reader what analysis was completed and how to interpret the values.
- The Discussion does not include an implications section, and the results are not discussed in context of existing/emerging research on athletes and mental health.
- The references are not current and are presented in various formatting styles. Come references are incomplete (e.g., reference 5).
Author Response
Please find attached our responses. Thank you for your consideration of this revised manuscript.

Round 2
Reviewer 1 Report
Thank you for the opportunity to review this revised manuscript. I would like to thank the authors for their responses to my initial review. I believe this manuscript requires further revision and my comments are included below. Some of these have emerged in this revised manuscript, while others I believe were either missed or not sufficiently addressed from my previous review. I would request that the authors please attend to these comments and resubmit as this is an interesting paper with potential interest to many readers.
Major comments:
- I appreciate the authors inclusion of their core hypothesis (that AI will be associated with symptoms of anxiety and depression). I would request the authors also include hypotheses related to their other analysis, specifically regarding the relationships between demographic variables, AI, and symptoms of anxiety and depression.
- I am primarily concerned that the authors have not adequately revised their discussion and risk misrepresenting the findings of this study without adequate discussion of appropriate literature. My specific notes are:
- I am curious about the author's paragraph on the different rates of anxiety and depression across high school and college students. There is quite a lot of evidence exploring this, and it is usually found that rates of anxiety and depression increase with age over teen years and young adulthood (see https://www.sciencedirect.com/science/article/pii/S0140673622010121). I would recommend the authors discuss how their findings relate to past research, as this provides weight for the suggestion that perceptions about control under COVID may have played a role here.
- I feel resistant to the authors phrase suggesting we need more evidence to understand if there is a 'cause and effect' relationship between depression and suicide. There has been extensive research on this in the field of suicide research, and while it is clear that depression and suicide are related this notion of a causative relationship has long been rejected as oversimplified. I would suggest reviewing literature such as this https://www.ncbi.nlm.nih.gov/pmc/articles/PMC7113180/ discussing the complex relationship between depression and suicide. If the authors still wish to include this sentence, I would ask that the authors please justify their continued inclusion of this phrase. The remainder of this paragraph on the link between depressive symptoms and suicide is acceptable, but I would suggest the authors link this more strongly to their present findings. I assume this paragraph is included because the authors found higher rates of depression in their sample, and they're concerned that this could relate to suicide risk. I would suggest the authors make this link explicit and perhaps note that we do not know if the students in their sample experienced suicidal thoughts and behaviors.
- The authors' paragraph on athletic identity findings seems to continue to suggest that the COVID pandemic was the mechanism by which depression symptoms were higher for the high AI group, e.g. "The findings that student athletes with higher AI scores had higher symptoms of depression underscores the detrimental effects of the COVID pandemic on athlete's mental health." I would advise the authors to revise this. This finding does not reasonably 'underscore this'. The present findings suggest that during the COVID pandemic student athletes with higher AI scores were more likely to have depression symptoms than their peers with lower AI. We could hypothesize that this is related to student athletes with higher AI being more impacted by the detrimental impacts of COVID on sport, but this is not clearly supported by the findings.
- The authors note that some reasons why anxiety might be different within groups of their study population could include "sudden loss of athletic training, decreased social interaction with peers, a sudden shift in academic platforms, from in-person learning to virtual, and a more generalized uncertainty regarding a possible end to the pandemic. " Why would these vary across race?
- The authors report both in their response to my review and earlier in the manuscript that a portion of students did not return their surveys (16.2%) and another portion were excluded for incomplete data (1.9%). However, in the limitation section the authors report that 18% of participants who submitted a survey were excluded. Which is the correct number? Please report this consistently.
Minor comments:
- The abstract seems very long (500+ words). I would advise the authors abbreviate this to be closer to a conventional abstract length (200 words). There is no need to report all specific findings in this abstract, and the authors may wish to present a briefer summary of findings.
- In the introduction, the authors have included a new paragraph describing research on mental health for athletes completed during the COVID pandemic. While the description of the first study (ref 21) is clear, the summary of the second (ref 22) is quite hard to follow. I would suggest the authors revise their summary of this study's findings to be clearer to follow and relate to the present research.
- The subsequent paragraph in the introduction begins with the phrase 'These statistics...' To which statistics is this comment referring? I would request the authors please clarify this.
- Throughout this revised manuscript, there are some minor grammatical errors and word repetition which appear related to the process of revising the manuscript. Some examples include in the PHQ-4 section where I noted the following issues 1) The phrase 'geared targeting depression' which I believe should read 'targeting depression'; 2) Repetition of content, the sentence '...that elucidates two items unique to anxiety and two items geared targeting depression [29].' Is redundant given the following sentence 'The first two questions in this section focus on anxiety'; 3) The sentence "participants were considered to have anxiety if the sum of their scores for A1 + A2 was > 3 and depression if the sum of their scores for D1 and D2 was over 3" is now redundant given the subsequent paragraph. I also noted repetition in the discussion, with the sentence "AI also differed by gender, race...etc." seeming to repeat the content of the previous sentence.
- I am curious about the authors' use of the term Caucasian as a race classifier. Was this the terminology included on the survey? I note that the authors use White in the abstract, which would be more appropriate. If Caucasian was the term the authors used on their survey, while they should include in this study for consistency with the survey, I would advise the authors review literature on the use of this term prior to conducting other studies including demographic self identification. For an introduction, I think this brief overview from the University of Minnesota summarizes the issues well https://med.umn.edu/news-events/time-phase-out-caucasian
- I would suggest the authors review their table titles. Table 2's title does not seem sufficiently descriptive. I would suggest the authors please revise this title to more fully capture the findings included in this table, which appears to reflect the spread of low/medium/high athletic identity across a range of participant characteristics. I feel similarly about the title of Table 4. Similarly, Table 3's title seems to suggest that race, ethnicity, school level etc. are the determinants of athletic identity and symptoms of anxiety and depression. This table shows a relationship or association between these variables, not determinants.
- Please remove the statistics from paragraph 1 of the discussion. In the first paragraph of the discussion, it is conventional to include a brief description or summary of findings without statistics.
- In the discussion, the authors use the word 'trends' to describe their findings. Given that this study is cross-sectional, I would request that the authors change this word choice for clarity that trends over time cannot be observed from this study.
- Please consider your use of language around race, e.g., 'Blacks' 'Asians' 'Caucasians'. Replacing these words with phrases like 'Black participants' and 'Asian people' would be more appropriate. I would recommend reviewing the issues with these terms, this video could be a starting point https://www.youtube.com/watch?v=QQ0hCEDH-aU
Author Response
Thank you so much for consideration of this revised manuscript. Please find attached responses to your comments.
